# Disulfide modification and thiol protection via tris(trimethylsilyl)silane-mediated hydrosilylation of disulfides

Ying Zhang[1], Kejun Lin[1], Zhenming Zang[1], Jianhui Chen [2] & Tingshun Zhu [1]✉

Thiol functionalities are indispensable in both biological and synthetic chemistry. However, unlike the powerful silylation strategies for hydroxyl groups, widespread applications in sulfur chemistry are severely hampered by the weak and hydrolytically labile nature of the Si−S bond. Here, we report that tris(trimethylsilyl)silane (TTMSS), a unique trisilyl-substituted reagent, efficiently enables the rapid hydrosilylation of disulfides under exceptionally mild conditions. This reaction affords robust silyl sulfides that show significantly enhanced hydrolytic stability compared to conventional analogues. The robust platform enables a practical, readily available strategy for orthogonal thiol protection and late-stage disulfide modification in complex molecules. Crucially, this finding reveals that the kinetic and thermodynamic properties of the Si−S bond can be finely tuned by strategic silyl substitution, establishing a general principle that reinvigorates silyl-protective methods for challenging sulfur chemistry.

Sulfur-containing functional groups play pivotal roles in biology, maintaining protein structures, regulating enzymatic activity, and mediating cellular redox homeostasis[1–4]. These versatile functions have inspired growing interest in the selective chemical manipulation of sulfur-containing motifs in peptides and small molecules. Among them, thiol and disulfide groups are particularly attractive handles for reversible and orthogonal transformations relevant to chemical biology[5–10]. However, the high susceptibility of thiols to oxidation and undesired disulfide formation necessitates protection under mild and orthogonal conditions. In solid-phase peptide synthesis (SPPS), cysteine thiols are commonly protected by trityl (Trt), acet-amidomethyl (Acm), methoxybenzyl (Mob), tert-butyl (tBu), and S-tert-butyl (StBu) groups[11–13]. However, many of these deprotection methods rely on heavy-metal reagents such as mercury, silver, or iodine oxidants, or involve strongly acidic or reductive conditions that are incompatible with sensitive substrates. The development of new strategies that enable mild, metal-free, and orthogonal deprotection has therefore become highly valuable for modern synthetic and chemical biology applications.

Silyl protecting groups have been widely employed on hydroxyl groups[14,15]. Using fluorides as unique removal conditions, the silyl protecting group provides an orthogonal strategy to other protection methods. Besides, by tuning the steric bulk of the substituents on silicon, different silyl ethers can be sequentially cleaved, enabling precise functional group control in multistep synthesis[16]. Inspired by these successes, several attempts have been made to extend silyl protecting groups in thiol chemistry[17,18]. However, in comparison to the stable Si−O bond (-110 kcal mol$^{-1}$), the Si−S bond is much weaker (-80 kcal mol$^{-1}$), and even weaker than the common Si−H bond (-95 kcal mol$^{-1}$) (Fig. 1a)[19,20]. These thermodynamic facts not only make the formation of the Si−S bond difficult but also pose a greater challenge to its stabilization against hydrolysis. Becker and co-workers first demonstrated a thermal reaction of diphenyl disulfide with pentaphenyl disilane at 190 °C to afford silyl sulfides in 1988[21]. More recently, Hreczycho and colleagues developed a similar Ru-catalyzed reaction of triethylsilane and disulfides at 120 °C using the Ru catalyst[22]. Brook and coworkers developed a B(C$_6$F$_5$)$_3$-catalyzed hydrosilylation[23,24] of disulfides using hydrosilanes and silicones (Fig. 1b). Dehydro-cross-

[1]Key Laboratory of Bioinorganic and Synthetic Chemistry of Ministry of Education, Guangdong Provincial Key Laboratory of Chiral Molecule and Drug Discovery, School of Chemistry, IGCME, Sun Yat-Sen University, Guangzhou, PR China. [2]Division of Gastrointestinal Surgery Center, the First Affiliated Hospital of Sun Yat-sen University, Guangzhou, Guangdong, PR China. ✉e-mail: zhutshun@mail.sysu.edu.cn

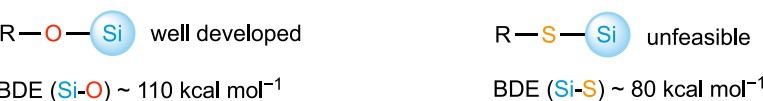

**(a) Silyl protecting groups for O and S**

R—O—Si  well developed

R—S—Si  unfeasible

BDE (Si-O) ~ 110 kcal mol$^{-1}$

BDE (Si-S) ~ 80 kcal mol$^{-1}$

**(b) Previous works for thiol silylation**

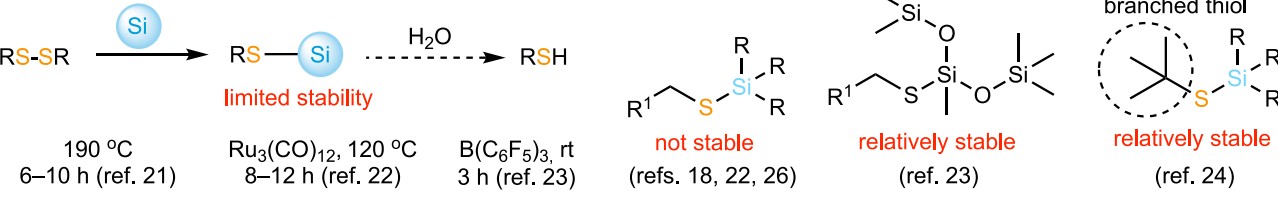

RS-SR $\xrightarrow{\text{Si}}$ RS—Si $\dashrightarrow{\text{H}_2\text{O}}$ RSH

limited stability

190 °C
6–10 h (ref. 21)

Ru$_3$(CO)$_{12}$, 120 °C
8–12 h (ref. 22)

B(C$_6$F$_5$)$_3$, rt
3 h (ref. 23)

not stable
(refs. 18, 22, 26)

relatively stable
(ref. 23)

branched thiol

relatively stable
(ref. 24)

**(c) The feature of (TMS)$_3$SiH**

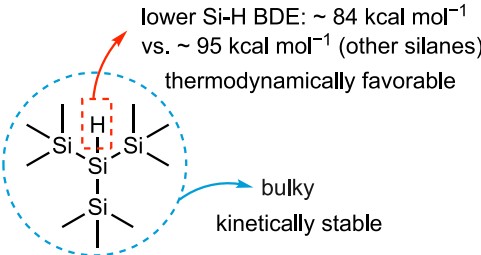

lower Si-H BDE: ~ 84 kcal mol$^{-1}$
vs. ~ 95 kcal mol$^{-1}$ (other silanes)
thermodynamically favorable

bulky
kinetically stable

**(d) This work**

R—S—S $\xrightarrow[\text{35 °C, 5 min}]{\text{(TMS)}_3\text{SiH}}$ R—S—Si(TMS)$_3$
or with 450 nm light ... SH

✓ Robust reaction under mild conditions

✓ Silyl protecting group for thiol chemistry

✓ Late-stage disulfide modification

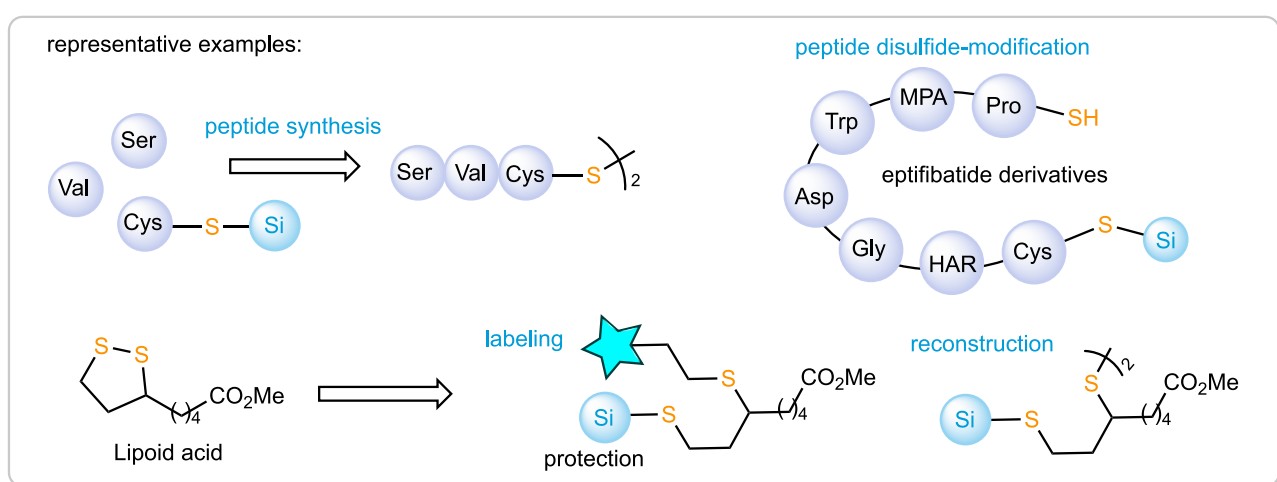

representative examples:

peptide synthesis

Val / Ser / Cys—S—Si → Ser Val Cys—S$_2$

peptide disulfide-modification

Trp / MPA / Pro—SH
Asp
Gly / HAR / Cys—S—Si

eptifibatide derivatives

Lipoid acid

labeling

protection
Si—S ... S ... CO$_2$Me

reconstruction
Si—S ... S$_2$ ... CO$_2$Me

**Fig. 1 | TTMSS mediated disulfide-hydrosilylation. a** Comparison of Si–O and Si–S silyl protection. **b** Limitations of previous thiol silylation methods. **c** Key features of (TMS)$_3$SiH. **d** This work: Mild and rapid thiol silylation for robust S–Si formation and late-stage disulfide modification.

couplings of silanes and thiols mediated by organometallic catalysts[25–30], as well as radical-mediated reactions of hydrosilanes with sulfur-containing substrates, have also been achieved[31]. Efforts have also been made to construct the Si–S bond from Si–Cl, Si–N, and Si–C precursors[18,32–35]. However, in many cases, thiolsilanes with traditional silyl protecting groups are nomally moisture-sensative and easily-decomposed during purification on silica gel[18,22–24,26]. Therefore, until very recently, the silyl protecting group strategy remained practically unviable for general use in thiol chemistry.

Among the silyl protecting agents, tris(trimethylsilyl)silane (TTMSS) is notable for its water and oxygen tolerance, high stability, and low toxicity due to its large steric hindrance[36,37]. Compared with other silanes, it has a more active Si–H bond (BDE ~ 84 kcal mol$^{-1}$, vs. ~95 kcal mol$^{-1}$ for other silanes) to generate silyl radical and bulkier substituents to stabilized the silyl radical (Fig. 1c)[17]. These features

make TTMSS particularly valuable in radical chemistry across organic synthesis, polymer science, and materials chemistry[37]. The thermodynamically active Si–H bond and the kinetic stabilization effect of TTMSS also motivate our exploration of its use for thiol protection. Herein, we found that TTMSS, owing to its thermodynamically active Si–H bond, enables rapid hydrosilylation of disulfides under mild conditions. More importantly, due to the kinetic stabilization effect of the bulky tris(trimethylsilyl)silyl group, the S–Si(TMS)$_3$ compounds show remarkably higher stability against hydrolysis than previously reported S–Si compounds. At the same time, this protecting group can be readily removed with fluoride ion. This approach provides a robust platform with consistently high yields and demonstrates broad applicability, ranging from thiol-group protection in the synthesis of cysteine-containing peptides to late-stage modifications of disulfide-containing functional molecules (Fig. 1d). As representative examples,

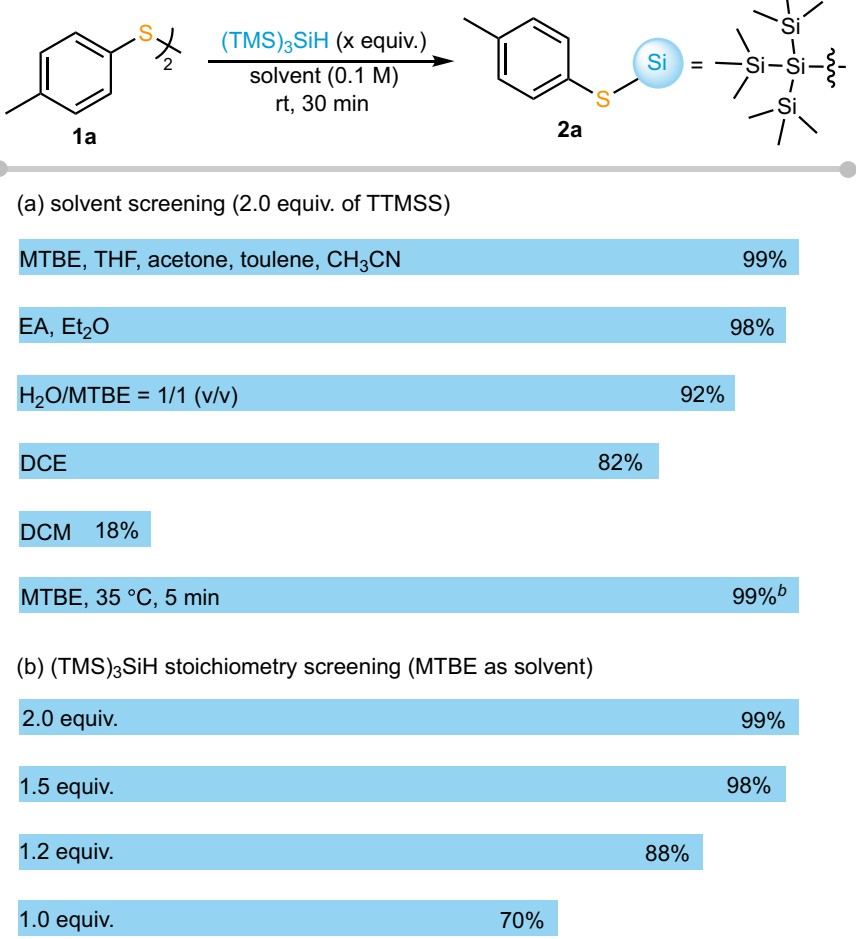

(a) solvent screening (2.0 equiv. of TTMSS)

| | |
|---|---|
| MTBE, THF, acetone, toulene, CH₃CN | 99% |
| EA, Et₂O | 98% |
| H₂O/MTBE = 1/1 (v/v) | 92% |
| DCE | 82% |
| DCM | 18% |
| MTBE, 35 °C, 5 min | 99%[b] |

(b) (TMS)₃SiH stoichiometry screening (MTBE as solvent)

| | |
|---|---|
| 2.0 equiv. | 99% |
| 1.5 equiv. | 98% |
| 1.2 equiv. | 88% |
| 1.0 equiv. | 70% |

**Fig. 2 | Optimization of TTMSS-mediated S–Si bond formation[a]. a** Solvent screening. **b** TTMSS stoichiometry screening. [a]Conditions: **1a** (0.10 mmol), rt, 30 min. [b]solvent removal in a rotary evaporator, 35 °C, 5 min. The yields were directly measured in ¹H NMR using 1,3,5-trimethoxybenzene as the internal standard. MTBE = 2-methoxy-2-methylpropane.

the method is applied to cysteine peptide synthesis, peptide disulfide-modification, and the late-stage modification of the drug lipoid acid involving regioselective thiol protection along with fluorescence labeling or disulfide reconstruction.

## Results and discussion
### Method optimization
The hydrosilylation reaction can be initiated by simply mixing TTMSS with disulfides in solvents. As shown in Fig. 2a, this method was compatible with a wide range of solvents, achieving quantitative yield in MTBE, THF, acetone, toluene, acetonitrile, ethyl acetate, and diethyl ether, and with high efficiency also in aqueous media (92% yield). Chlorinated solvents were not very compatible with this hydrosilylation reaction and led to a significant decrease in yield (82% yield in dichloroethane and 18% yield in dichloromethane). Further study showed that the rapid reaction can be accomplished during the solvent removal process in a rotary evaporator at 35 °C for 5 min, affording **2a** quantitatively. Stoichiometric studies also showed the simplicity of the reaction (Fig. 2b). Increasing the equivalents of TTMSS from 1 equiv. to 2 equiv. remarkably accelerated the reaction and increased the yield from 70% to quantitative conversion. In all cases, thiols were generated as byproducts and the dehydrogenative S–Si coupling between thiol and TTMSS was not observed (see Supplementary Fig. 2 for details). Attempts to re-oxidize the thiol byproduct to the corresponding disulfide for full conversion to the thiosilane were unsuccessful (see Supplementary Table 2 for details).

### Substrate scopes
With the optimized conditions in hand, we first tested the substrate scope of diaryl disulfides, and the results are shown in Fig. 3. Most aryl disulfides underwent complete hydrosilylation simply by rotary evaporation at 35 °C for 5 min, and clean reaction systems with only products and starting reagents were observed in all cases. Aromatic disulfides bearing electron-donating or electron-withdrawing substituents at the *para* position of the aryl ring all reacted rapidly to give the desired silylation products (**2a**–**2i**). Take product **2g** as a representative example, in-situ ¹⁹F NMR monitoring revealed the kinetic features of the reaction. In contrast to the rapid transformation under rotavap conditions (99% yield within 5 min), the conventional closed-system reaction at a concentration of 0.1 M was much slower (88% yield after 24 h). The solvent-free reaction also proceeded slowly (95% yield after 30 h), probably due to the poor solubility of the substrate in TTMSS. As the steric hindrance on the aromatic ring increased (**2j**–**2l**), the yields were essentially unaffected. The fully fluorinated disulfide, featuring a strong electron-withdrawing effect, also exhibited excellent reactivity, affording the desired product **2m** in 93% yield. The structure of **2m** was unambiguously confirmed by single-crystal X-ray diffraction. While most reactions proceeded smoothly in methyl tert-butyl ether, the less polar 1,2-di(naphthalen-2-yl)disulfide required toluene as the solvent, giving **2n** in 78% yield.

Apart from diaryl disulfides, dialkyl disulfides are also viable substrates. However, due to the stronger S–S bond in dialkyl disulfides (e.g., BDE 51.8 kcal mol⁻¹ for **1a** vs. 67.4 kcal mol⁻¹ for **1o**)[38], some

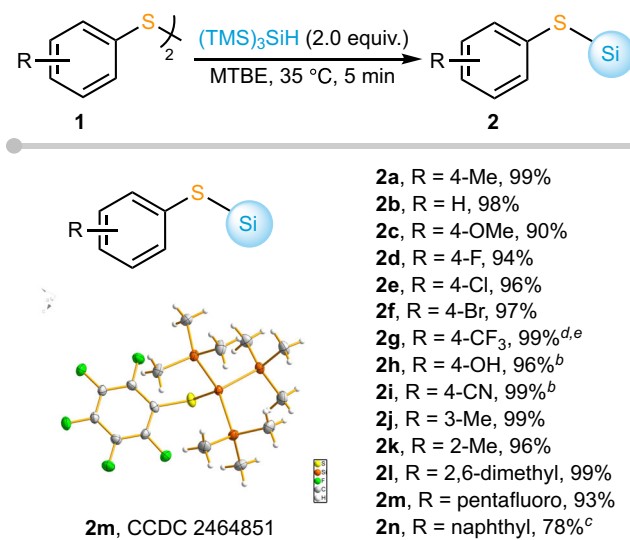

**2a**, R = 4-Me, 99%
**2b**, R = H, 98%
**2c**, R = 4-OMe, 90%
**2d**, R = 4-F, 94%
**2e**, R = 4-Cl, 96%
**2f**, R = 4-Br, 97%
**2g**, R = 4-CF$_3$, 99%[d,e]
**2h**, R = 4-OH, 96%[b]
**2i**, R = 4-CN, 99%[b]
**2j**, R = 3-Me, 99%
**2k**, R = 2-Me, 96%
**2l**, R = 2,6-dimethyl, 99%
**2m**, R = pentafluoro, 93%
**2n**, R = naphthyl, 78%[c]

**2m**, CCDC 2464851

**Fig. 3 | Substrate scope of aryldisulfides[a].** [a]The reactions were performed with **1** (0.10 mmol), TTMSS (2.0 equiv.), MTBE, 35 °C, 5 min, solvent removal in rotary evaporator. Isolated yields are shown. [b]6 h. [c]toluene as the solvent instead of MTBE, 6 h. [d]closed system, 0.1 M, 24 h, 88% yield. [e]solvent-free, 30 h, 95% yield.

substrates required an increased amount of TTMSS or 450 nm photoirradiation to achieve optimal yields (see Supplementary Tables 3, 4 for for optimization details, Supplementary Table 6 for challenging substrates, and Supplementary Fig. 1 for the irradiation setup). As shown in Fig. 4, the reaction was compatible with a wide range of functional groups, including alcohols, carboxylic acids, alkynes, alkenes, esters, ketals, amides, ketones, and various heterocycles.

The benzylic disulfide reacted efficiently to afford the corresponding silylthioether **2o** in 82% yield, and its structure was unambiguously confirmed by single-crystal X-ray diffraction. Primary, secondary, and tertiary alkyl disulfides all reacted efficiently under the standard conditions, affording the corresponding silylthioethers **2p**–**2t** in excellent yields. The molecular structure of **2t** was confirmed by single-crystal X-ray diffraction. Furthermore, the reaction also tolerated heteroaryl substrates, and the difurfuryl disulfide bearing electron-rich furan rings delivered the desired product **2u** in 99% yield. Even the less stable acyl disulfide underwent smooth conversion, affording the desired product **2v** in 76% yield. A series of amino acid–derived cystine disulfides bearing different protecting groups, including carboxylic functionalities, reacted smoothly to afford the corresponding products **2w**–**2y** in high yields (86–94%). The molecular structure of **2w** was unambiguously confirmed by single-crystal X-ray diffraction. A cyclic disulfide featuring a highly planar structure was also compatible under the standard conditions, affording the corresponding product **2z** in 94% yield. Oxidized DTT derivatives bearing various oxygen substituents (hydroxyl, acetyl, benzyl ether, and acetonide) were also well tolerated, affording the corresponding products **2z1**–**2z4** in high yields, and the molecular structures of **2z1** and **2z2** were confirmed by single-crystal X-ray diffraction. In addition, a sterically more congested cyclic disulfide bearing a free carboxylic acid group was also compatible, delivering the corresponding product **2z5** in 73% yield.

Building on the successful cleavage of aliphatic disulfides, we next extended this method to more complex, biologically relevant substrates, including lipoic acid derivatives and peptide-based disulfides, to assess its broader applicability. Unmodified lipoic acid was readily converted into the corresponding silyl sulfide **2z6** in good yield. The terminal alkyne-functionalized lipoic acid derivative also underwent smooth transformation to afford **2z7** in excellent yield, providing a

versatile handle for subsequent click-type reactions. Furthermore, a diverse range of lipoate esters (**2z8**–**2z17**) was efficiently transformed under the standard conditions, giving the corresponding silyl sulfides in good to excellent yields. These esters, derived from diterpenol (**2z9**), monoterpenols (**2z10**–**2z12**), indole alcohols (**2z13**), sugar alcohols (**2z14**), alkaloid-derived alcohols (**2z15**), and steroid-derived alcohols (**2z16, 2z17**), all reacted smoothly, further highlighting the broad substrate compatibility and potential utility of this protocol for biomolecule modification. In all cases, the hydrosilylation reactions proceeded with excellent regioselectivities (>20:1 r.r.), affording S−Si(TMS)$_3$ with the less-steric sulfur atoms while releasing the more-steric sulfur atoms as free thiol groups. Finally, to further demonstrate the applicability of this protocol to structurally complex biomolecules, we applied it to the cyclic peptide drug eptifibatide, which bears a single disulfide bond and is clinically used as an antiplatelet agent for the treatment of acute coronary syndromes. The corresponding silyl sulfide **2z18** was obtained in 50% yield, with approximately 50% of starting eptifibatide unreacted, under clean reaction conditions, highlighting the potential of this method for late-stage functionalization of biologically active peptides.

## Synthetic applications

The practicality of this method was demonstrated by scaling up the reaction to a 5 mmol scale, which afforded the target product **2x** in 90% yield. The Boc group of **2x** was subsequently removed using TMSOTf/NMM, without affecting the S−Si(TMS)$_3$ protecting group, to give **2z19** in 60% isolated yield (The declined yield is due to the side reaction of the amine group and ester group). This experiment demonstrates the orthogonal compatibility of the silyl protecting group with other protecting strategies (Fig. 5a). The stability of S−Si(TMS)$_3$ was tested using **2a** as a model substrate. The control experiments under aqueous acidic conditions show that the S−Si(TMS)$_3$ group is much more stable than the O-TMS group and slightly more stable than the O-TBS group (see Supplementary Table 5 for details). To illustrate the utility of silyl-protected cysteine in peptide synthesis, dipeptide **2z20** was subjected to HATU-mediated condensation with silyl-protected cysteine, affording the tripeptide **2z21**. The tripeptide was then converted to the disulfide-bridged tripeptide **2z22** via sequential TBAF-promoted desilylation and H$_2$O$_2$ oxidation (Fig. 5b)[39]. Using lipoic acid derivatives as model substrates, we further demonstrated the late-stage disulfide modification. As shown in Fig. 5c, a disulfide-rebridged dimer of lipoic acid ester **2z23** was obtained in high yield after hydrosilylation and subsequent disulfide formation, highlighting the potential in disulfide rebridging[5,40,41] and peptide biohybrid construction[42–44]. The post-synthetic diversification with other click reactions was also evaluated. As shown in Fig. 5d, after a regioselective hydrosilylation of the alkyne-containing lipoic acid derivative **1z7**, the exposed thiol group in **2z7** underwent thiol-Michael addition with benzyl acrylate to give **2z24** in 85% yield[45,46], followed by copper(I)-catalyzed azide-alkyne cycloaddition (CuAAC) with benzyl azide to afford **2z25** in 83% yield[47,48]. Thiol-Michael addition of pyren-1-ylmethyl acrylate to **2z7** afforded adduct **2z26** in 65% yield, converting the non-emissive substrate into a fluorescent derivative and suggesting that TTMSS-mediated disulfide hydrosilylation may provide a controllable approach to disulfide labeling relevant to biomolecular and redox-responsive systems[8,49].

## Mechanistic studies and proposed mechanism

To study the mechanism of the hydrosilylation reaction, a series of control experiments was performed. Radical trapping with 2,2,6,6-tetramethylpiperidine-1-oxyl (TEMPO) markedly inhibited the reaction, affording only 14% yield. The corresponding adduct (ArS-TEMPO) was detected by high-resolution mass spectrometry (HRMS)[50], supporting the involvement of radical intermediates (see Supplementary Information for details). A spin-trapping experiment with PBN under

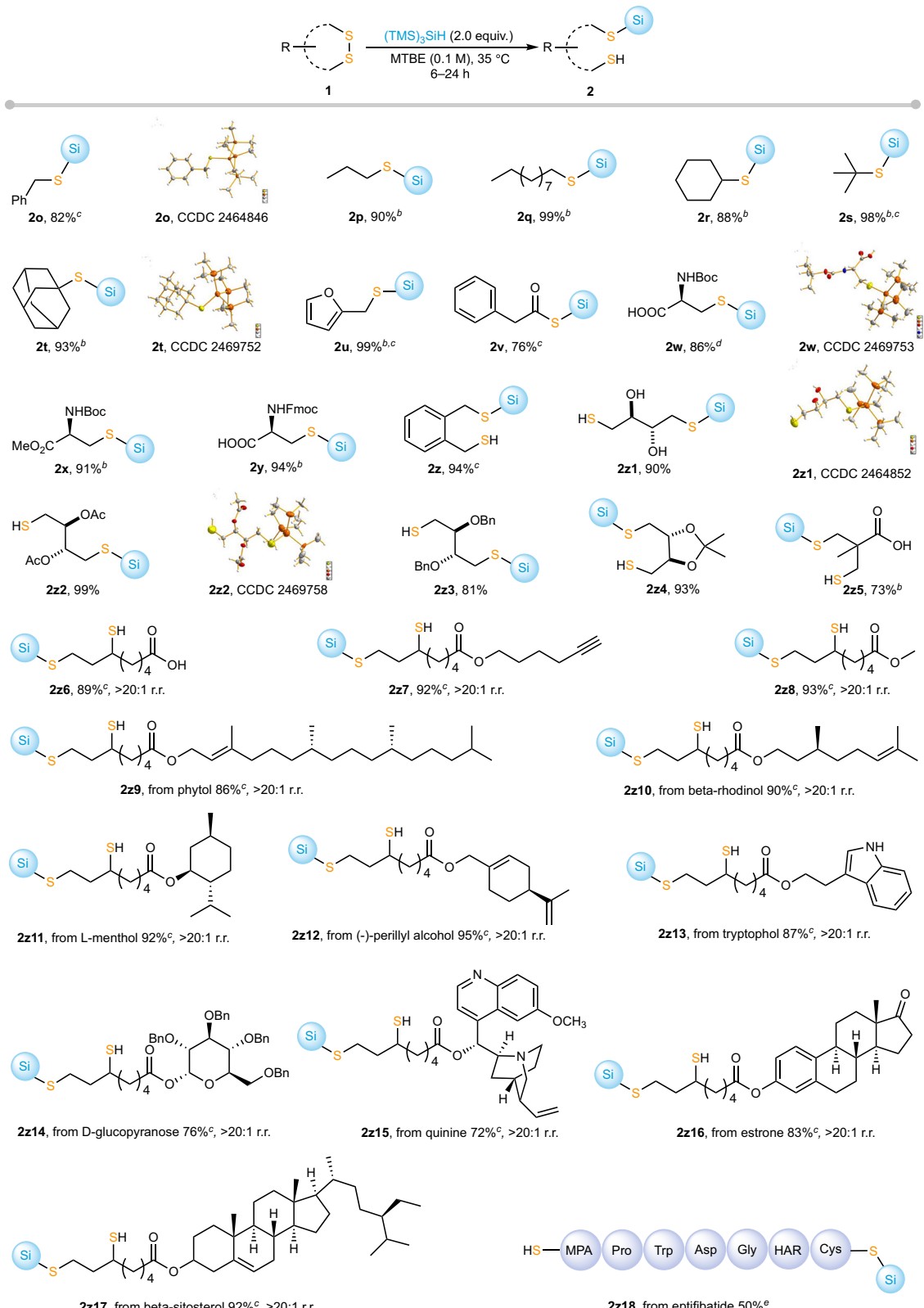

**Fig. 4 | Substrate scope of alkyldisulfides**[a]. [a]The reactions were performed with **1** (0.10 mmol), TTMSS (2.0 equiv.), MTBE (0.1 M), 35 °C, isolated yields are shown. [b]450 nm, 6 W, rt. [c]TTMSS (4.0 equiv.) was used. [d]MTBE (0.5 M) was used. [e]TTMSS (100 equiv.), DMF/MTBE = 3/1 (v/v), 0.006 M, 450 nm, 6 W, 24 h, rt, HPLC yield. r.r. regioisomeric ratio.

visible-light irradiation (450 nm) revealed a characteristic signal of a PBN-thiyl radical adduct upon irradiation, confirming the involvement of sulfur-centered radical intermediates (Fig. 6a)[51]. In addition, kinetic isotope effect (KIE) measurements gave a $k_H/k_D$ value of 8.7,

suggesting that cleavage of the Si–H bond is involved in the rate-determining step (Fig. 6b). We propose that the reaction is initiated via preferential formation of thiyl radicals rather than silyl radicals, because the homolytic S–S bond dissociation energy

**(a) Gram-scale reaction and Boc deprotection**

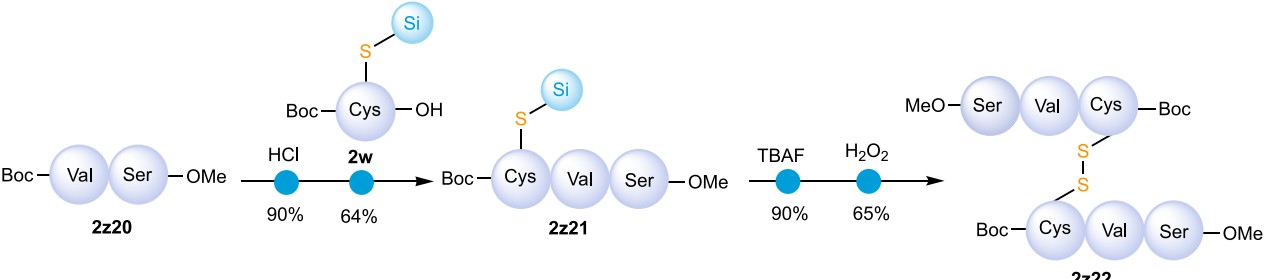

**(b) Peptide synthesis via TTMSS-protection of Cys**

**(c) Dimerization of silicon-protected lipoic acid derivatives**

**(d) Multistep click reactions**

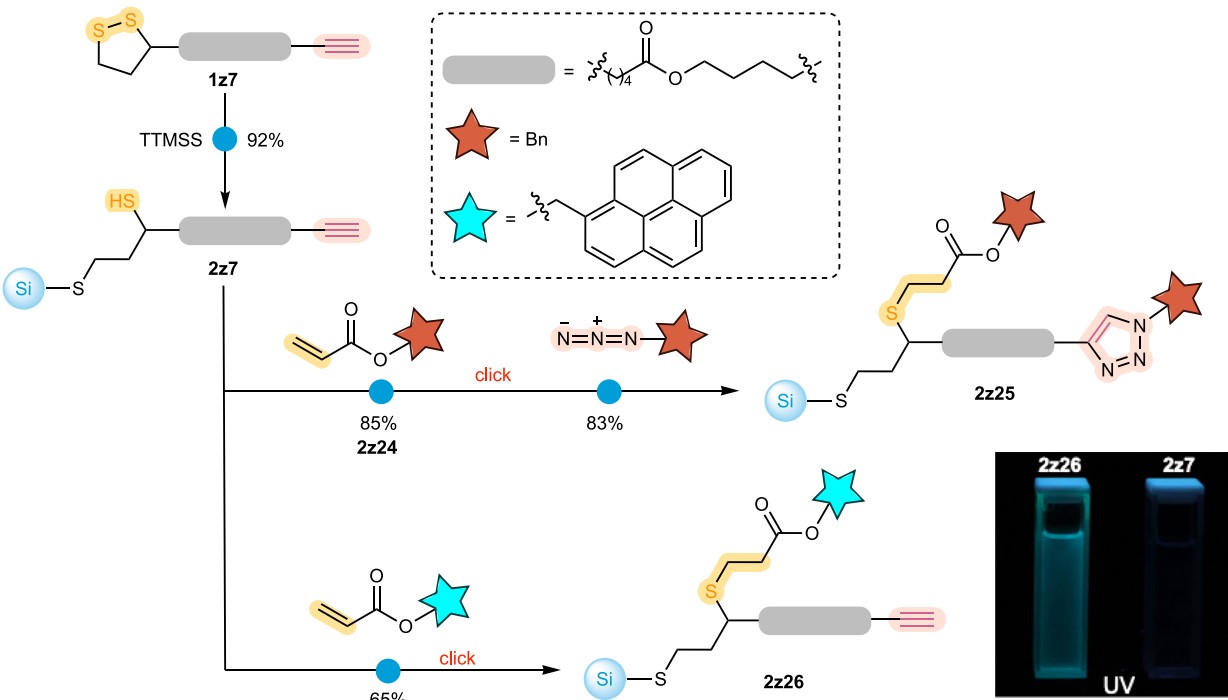

**Fig. 5 | Gram-scale synthesis and product transformations. a** Gram-scale synthesis and Boc deprotection of **2x**. **b** Peptide synthesis using TTMSS-protected cysteine. **c** Dimerization of silicon-protected lipoic acid derivatives. **d** Multistep click reactions of silyl-protected disulfides. Reaction conditions: **a** (TMS)$_3$SiH (2.0 equiv.), MTBE (0.1 M), 450 nm, 30 W, 24 h; then TMSOTf (3.0 equiv.), NMM (3.0 equiv.), DCM, rt, 6 h. **b** HCl (4 M in 1,4-dioxane, 2 mL), 0 °C, 1 h; then **2w** (1.0 equiv.), HATU (1.1 equiv.), NMM (2.0 equiv.), DCM, rt, 12 h; followed by TBAF (1 M in THF, 0.5 mL, 1.0 equiv.), 0 °C, 20 min, and H$_2$O$_2$ (1.0 equiv.), NaI (1 mol%), EA, 0 °C, 20 min. **c** TTMSS (4.0 equiv.), MTBE (0.1 M), 35 °C, 24 h; then 2,2'-dithiodipyridine (DTDP, 0.5 equiv.), Cs$_2$CO$_3$ (1.5 equiv.), DCM, rt, 4 h. **d** Benzyl acrylate (1.1 equiv.), Cs$_2$CO$_3$ (1.0 equiv.), DCM (0.1 M), rt, 30 min; then BnN$_3$ (2.0 equiv.), CuTc (20 mol %), toluene (0.1 M), N$_2$, rt, 6 h.

**(a) Spin-trapping experiment**

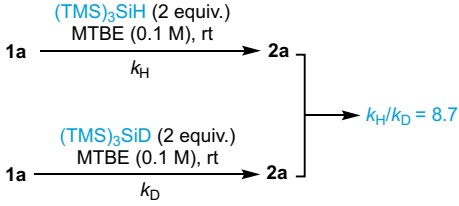

**(c) Fluorescence spectra of TTMSS and cyclohexyldisulfide**

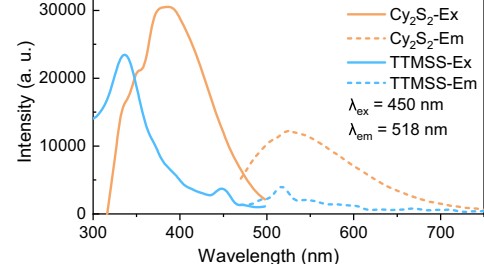

**(b) Kinetic isotope effect (KIE) experiment**

**(d) Proposed mechanism**

Fig. 6 | **Proposed mechanism with supporting evidence. a** EPR spin-trapping experiment showing the PBN-thiyl radical adduct (blue line: experimental; red line: simulated). Radical **1a-1**: g = 2.0053, $a_N$ = 14.8 G, $a_H$ = 5.3 G. **b** Kinetic isotope effect (KIE) experiment with (TMS)$_3$SiH and (TMS)$_3$SiD ($k_H/k_D$ = 8.7). **c** Fluorescence spectra of TTMSS and cyclohexyldisulfide (Cy$_2$S$_2$) in MTBE ($\lambda_{ex}$ = 450 nm, $\lambda_{em}$ = 518 nm). Solid lines denote excitation (Ex) spectra and dashed lines denote emission (Em) spectra. Intensity is given in arbitrary units (a. u.). **d** Proposed reaction mechanism.

(-50–65 kcal mol$^{-1}$)[38], is much lower than that of the Si−H bond (-84–104 kcal mol$^{-1}$). Disulfides with lower BDEs can generate thiyl radicals even at room temperature[52], whereas those with higher BDEs require heating or photoirradiation to undergo homolysis, consistent with the light-promoted nature of this reaction[53]. To clarify why 450 nm light is required to promote the reaction, fluorescence spectra of TTMSS and cyclohexyldisulfide (Cy$_2$S$_2$) were recorded in MTBE ($\lambda_{ex}$ = 450 nm, $\lambda_{em}$ = 518 nm), as shown in Fig. 6c. Under 450 nm excitation, TTMSS displayed two excitation bands centered at 333 nm and 450 nm, while Cy$_2$S$_2$ showed a strong excitation peak at 384 nm with a broad absorption tail extending to 500 nm. These results indicate that both TTMSS and the disulfide absorb in the blue-light region, supporting that 450 nm irradiation effectively photoactivates the system and facilitates hydrosilylation. Based on these experimental observations, we propose two possible mechanistic pathways (Fig. 6d). In the stepwise pathway, homolytic cleavage of the disulfide generates thiyl radicals, one of which abstracts a hydrogen atom from TTMSS to form a silyl radical that subsequently couples with the second thiyl radical to yield the product. Alternatively, in a concerted pathway, the disulfide and TTMSS undergo a synchronous hydrogen transfer and electron reorganization via a four-membered transition state (TS1), generating both the thiol and silyl sulfide in a single step.

In conclusion, we have developed a clean and practical disulfide hydrosilylation that employs TTMSS as an efficient reagent for silylation-based thiol protection. The reaction proceeds under mild, metal-free conditions with broad substrate generality, affording stable silyl sulfides in consistently high yields. The simplicity, cleanliness, and robustness of this process make it readily applicable to complex molecular settings. In addition, this transformation provides a versatile platform for late-stage disulfide modification and orthogonal thiol protection, with potential applications in peptide synthesis and disulfide labeling. Overall, this study establishes a reliable and broadly applicable approach to Si−S bond construction and thiol protection, enabling precise and practical manipulation of sulfur-containing molecules in both synthetic and biological contexts.

## Methods

Representative procedure for the hydrosilylation of disulfides: A 4 mL glass vial equipped with a magnetic stir bar was charged with disulfide **1z1** (0.10 mmol) in MTBE (0.1 M). Tris(trimethylsilyl)silane (TTMSS, 2.0 equiv.) was added under air without inert-atmosphere protection. The reaction mixture was stirred at 35 °C for 12 h, and the reaction progress was monitored by TLC. After completion, the solvent was removed under reduced pressure, and the residue was purified by silica gel column chromatography to afford the corresponding silyl sulfide. Reaction conditions (e.g., TTMSS equivalents, irradiation at 450 nm, and reaction time) were adjusted for specific substrates when necessary. Detailed experimental procedures and characterization data are provided in the Supplementary Information.

## Data availability

The crystallographic data for compounds **2m**, **2o**, **2t**, **2w**, **2z1**, and **2z2** generated in this study have been deposited in the Cambridge Crystallographic Data Centre (CCDC) under deposition numbers 2464851, 2464846, 2469752, 2469753, 2464852, and 2469758. These data can be obtained free of charge from the CCDC via https://www.ccdc.cam.ac.uk/structures/. All other data supporting the findings of this study are available within the article and Supplementary Information. All data are available from the corresponding author upon request.

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

## Acknowledgements
We acknowledge financial support from the National Natural Science Foundation of China (No. 22071269), Pearl River Recruitment Program of Talent (No. 2019QN01L149), and Guangdong Provincial Key Laboratory of Construction Foundation (No. 2023B1212060022).

## Author contributions
Y.Z. performed the investigation, conducted formal analysis, curated the data and wrote the original draft. K.L. contributed to writing, review, and editing. Z.Z. performed the X-ray crystallographic analysis. J.C. contributed to conceptualization and scientific discussion. T.Z. supervised the project, provided resources and contributed to conceptualization. All authors discussed the results and approved the final manuscript.

## Competing interests
The authors declare no competing interests.
