## [Transparent Peer Review file · Nature Communications]

Disulfide Modification and Thiol Protection via Tris(trimethylsilyl)silane-Mediated Hydrosilylation of Disulfides

Corresponding Author: Professor Tingshun Zhu

Version 0:

Reviewer comments:

Reviewer #1

(Remarks to the Author)

Zhu and coworkers developed a click-type reaction for Si-C bond formation with disulfide hydrosilylation using TTMS. More than 50 examples, including peptide synthesis and late-stage modifications, were provided to show the robustness of the reaction.

To be honest, the hydrosilylation of disulfide is an old reaction that has been studied since the 1980s. However, all of the literature results show that the fragile Si-S bond is nearly infeasible for thiol protection. In this work, the authors discover an amazing fact that the bulky tris(trimethylsilyl)silyl group can greatly improve the hydrolytic stability of the resulting S-Si species, addressing the long-standing limitation in silicon-based sulfur protection. At the same time, the reactivity is also much higher than the traditional disulfide hydrosilylations. The radical quenching test and KIE test showed that the cleavage of the Si-H bond to generate the silyl radical is the RDS in the reaction. The stability of the TTMS radical possibly leads to the amazingly high reactivity of the reaction. As representative examples, the reaction with diaryldisulfide can be accomplished during the solvent removal process within 5 min. This high reactivity enables the impressive wide application, such as late-stage modification of medicine derivatives, cysteine peptide synthesis, S-S bond labeling, S-S bond reconstruction, etc.

With no doubt, this work represents a significant development in thiol-protection chemistry. Learning from the importance of sulfur chemistry in chemical biology and material science, I believe this work can attract broad attention. I am delighted to recommend acceptance after some minor revision.

1. For the unsymmetrical disulfides in Figure 4, the regioselectivity information should be provided.
2. As mentioned by the author, the controllable sequential deprotection is an important advantage in silyl protective chemistry. What is the stability sequence among S-TTMS, O-TMS, and O-TBS? Some related control tests are recommended to further clarify the feature of S-TTMS.
4. There are several inconsistencies in the use of typographical symbols. Product ranges are written with both a hyphen and an en dash (e.g., 2a-2i vs 2j-2k). These symbols should be in uniform format. The manuscript should be double-checked carefully for these minor problems.
5. Page 8, "to give 2z19 in 60% isolated yield. (The declined yield is due to the side reaction of the amine group and ester group)." should be "... yield (.....)." Explanatory parenthetical remarks that pertain to the sentence should be placed before the full stop.

Reviewer #2

(Remarks to the Author)

This manuscript describes a tris(trimethylsilyl)silane-mediated hydrosilylation of disulfides to access tris(trimethylsilyl)silyl sulfides as robust and fluoride-labile protecting groups for thiols. The authors demonstrate broad substrate scope, including aryl and alkyl disulfides, amino acid and peptide derived disulfides, and lipoic acid derivatives, and they further showcase

several useful transformations and late-stage functionalizations. The manuscript is generally well organized and clearly written. The Supporting Information is also carefully prepared.

However, I have several concerns:

1. In the Introduction, the authors state:

“However, in all cases, the Si–S compounds are normally moisture sensitive and easily decompose during storage or column chromatography.”

This statement does not appear to be fully supported by the literature, including ref. 29 (Eur. J. Org. Chem. 2021, 2694–2700) cited by the authors. In that and related studies, the behavior of silyl protecting groups for thiol was shown to follow the same steric trend as classical silyl ethers, i.e., bulkier silyl groups generally give more stable S–Si bonds. For example, in ref 29, Me₂PhSi–SR was also reported to be comparatively stable.

In the present work, the authors employ an even more sterically demanding group (TMS)₃Si–, so it is in fact quite reasonable to expect enhanced stability relative to less hindered silyl groups.

In Fig. 1b, the authors summarize several previously reported strategies for forming Si–S bonds. Given its relevance, the method described in ref. 29 (Eur. J. Org. Chem. 2021, 2694–2700) should be explicitly incorporated into this summary, together with a brief indication of its advantages/limitations compared to the present TTMSS-mediated protocol.

2. In fact, the synthesis of tris(trimethylsilyl)silyl sulfides has already been reported, for example in Tetrahedron 72 (2016) 7764–7769, where (TMS)₃Si–SR type compounds are also accessed. The authors should also cite this reference.

3. In Fig. 2b, the yields obtained with different equivalents of TTMSS are shown. When a 1:1 ratio of RSSR to (TMS)₃SiH is used, the reported yield appears to reach ~70%. This is somewhat confusing. In the reaction, RSSR + (TMS)₃SiH → RS–Si(TMS)₃ + RSH, the stoichiometry and the definition of “yield” (with respect to which component?) should be unambiguously consistent.

Reviewer #3

(Remarks to the Author)

Zhu presents a study on the use of tris(trimethylsilyl)silane as a silylating reagent for the formation of silicon–sulfur bonds. In general, this area has not been extensively explored, primarily due to the high susceptibility of silicon–sulfur bonds to hydrolysis, which hampers both the synthesis and practical application of such compounds. However, the surprisingly straightforward nature of the method presented by Zhu makes this manuscript worthy of closer consideration. The scope of this transformation is truly outstanding.

As in the case of cyclic disulfides, I can see why the authors refer to this transformation as hydrosilylation of disulfides. However, Figure 2 makes it look more like a silylation. There is no information shown about the formation of a thiol. Does this mean that, from the disulfide used, one half of the RS groups goes to form a thiosilane while the other half becomes a thiol, or do we obtain only the thiosilane, with S-silylation also occurring? This question is related to another one: if a thiol is indeed formed, how do the authors separate it from the thiosilane? Is this isolation straightforward? The thiols formed would be rather difficult to separate using a rotary evaporator (despite their high volatility), and the process would certainly not be very safe for the personnel.

The authors describe a (accelerated) synthetic procedure consisting simply of mixing the reagents on a rotavap at 35 °C, where the solvent slowly evaporates, ultimately yielding the product. But what happens if this is carried out in a closed system at 35 °C? And what about performing the reaction at this temperature under solvent-free conditions?

For the sentence “It is worth noting that the phenol group, normally an active functional group for silylation, was well tolerated in our reaction (product 2h, 96% yield),” it would be useful to refer to some literature. Are there known cases of silylation of OH groups without a catalyst? If so, why is that not the case here? Regardless of the answer (yes or no), this should be backed up with appropriate references (either an original paper or a review).

Radical trapping by TEMPO. There are known studies showing that TEMPO is not a suitable reagent in the case of thiols. It simply enables the formation of disulfides, in this case regenerating the starting material. It would be worthwhile referring to this literature. Nevertheless, this does provide some confirmation of the proposed mechanism. However, it might be worth attempting a radical clock reaction, for example using aliphatic disulfides bearing a bromo substituent, or disulfides containing cyclopropyl groups.

I suggest a major revision.

Version 1:

Reviewer comments:

Reviewer #1

(Remarks to the Author)

The authors have responded thoroughly to the reviewers' comments and revised the manuscript accordingly. In its current form, the manuscript meets the standard for acceptance.

Reviewer #2

(Remarks to the Author)

The authors have revised the manuscript carefully in response to this reviewer' comments.

Reviewer #3

(Remarks to the Author)

The responses provided by the authors are satisfactory. I am genuinely surprised by the high stability of the obtained thiosilanes and by the possibility of their isolation using column chromatography. Apart from this, the work looks significantly improved.

I congratulate the authors on a job well done.

Tingshun Zhu
zhutshun@mail.sysu.edu.cn
Professor, School of Chemistry
Sun Yat-Sen University, Guangzhou, China

January 29, 2026

Referee #1 was supportive of acceptance and provided several constructive suggestions.

1. For the unsymmetrical disulfides in Figure 4, the regioselectivity information should be provided.

Response: The unsymmetrical disulfides shown in Figure 4 (**2z6–2z17**) were all obtained with excellent regioselectivity (>20:1 r.r.), The corresponding description has been added in the manuscript as follows: “In all cases, the hydrosilylation reactions proceeded with excellent regioselectivities (>20:1 r.r.), affording S-TTMS with the less-steric sulfur atoms while releasing the more-steric sulfur atoms as free thiol groups.”

2. As mentioned by the author, the controllable sequential deprotection is an important advantage in silyl protective chemistry. What is the stability sequence among S-TTMS, O-TMS, and O-TBS? Some related control tests are recommended to further clarify the feature of S-TTMS.

Response: We thank the reviewer for this important suggestion. Control experiments were carried out to test the stability of S-Si(TMS)₃, O-TMS, and O-TBS using *p*-tolylloxy silanes and *p*-tolylthio silanes as substrates. As shown below, the OTMS group was easily hydrolyzed within 30 min in MeOH with silica gel while the OTBS and S-TTMS groups were stable. The S-TTMS group showed more stability in aqueous HCl (1M or 4M) than OTBS. This Table is added as Table S5 in SI, and related descriptions were added in the manuscript: “The stability of S-Si(TMS)₃ was tested using **2a** as a model substrate. The control experiments under aqueous acidic conditions show that the S-Si(TMS)₃ group is much more stable than the O-TMS group and slightly more stable than the O-TBS group (see Table S5 in the Supplementary Information for details).”

Table S5. Stability comparison of OTMS, OTBS, and S-Si(TMS)₃ protecting groups.

Conditions	p -Tol-OTMS	p -Tol-OTBS	p -Tol-S-Si(TMS) ₃
Silica gel/MeOH, rt	30 min (0%)	24 h (>95%)	24 h (>95%)
HCl (1 M), 20% H ₂ O, rt	5 min (0%)	2 h (8%)	2 h (42%)
HCl (4 M), 20% H ₂ O, rt	5 min (0%)	10 min (0%)	10 min (6%)
TBAF (1 M in THF), rt	3 min (0%)	10 min (0%)	15 min (0%)

^aGeneral conditions: Substrate (0.05 mmol). For silica gel stability tests: silica gel (50 mg), MeOH (1.0 mL). For acidic conditions: HCl (1 M or 4 M in 1,4-dioxane, 1.0 mL) with H₂O (0.2 mL). For fluoride-mediated deprotection: TBAF (1 M in THF, 1.0 equiv.), THF (1 mL). Reactions were conducted at room temperature. The yields were isolated recovery yields. *p*-Tol = 4-methylphenyl, TMS = trimethylsilyl, TBS = *tert*-butyldimethylsilyl.

3. There are several inconsistencies in the use of typographical symbols. Product ranges are written with both a hyphen and an en dash (e.g., 2a-2i vs 2j–2k). These symbols should be in uniform format. The manuscript should be double-checked carefully for these minor problems.

Response: revised.

4. Page 8, “to give 2z19 in 60% isolated yield. (The declined yield is due to the side reaction of the amine group and ester group.)” should be “... yield (....)” Explanatory parenthetical remarks that pertain to the sentence should be placed before the full stop.

Response: revised.

Referee #2 provided several helpful comments aimed at more clarified description.

- In the Introduction, the authors state: “However, in all cases, the Si–S compounds are normally moisture sensitive and easily decompose during storage or column chromatography.” This statement does not appear to be fully supported by the literature, including ref. 29 (Eur. J. Org. Chem. 2021, 2694–2700) cited by the authors. In that and related studies, the behavior of silyl protecting groups for thiol was shown to follow the same steric trend as classical silyl ethers, i.e., bulkier silyl groups generally give more stable S–Si bonds. For example, in ref 29, Me₂PhSi–SR was also reported to be comparatively stable. In the present work, the authors employ an even more sterically demanding group (TMS)₃Si–, so it is in fact quite reasonable to expect enhanced stability relative to less hindered silyl groups.

Response: We thank the reviewer for this helpful comment. Indeed, the stability of S–Si linkages strongly depends on steric demand and reaction conditions. For some special steric-hindered substrate, as mentioned by the reviewer (Eur. J. Org. Chem. 2021, 2694–2700 (2021), ref. 29 was changed to ref. 23), the traditional silyl protecting group can also give a stable product. We have therefore revised the statement in the introduction to adopt more appropriate, literature-consistent phrasing. To make this point more transparent, we summarized representative literature observations in the table shown below, separating steric contributions from the silicon substituent and from the thiol fragment.

Representative structure	Sterics at Si	Sterics at S	Stability descriptor	Reference
PhS–SiEt ₃	Low	Low	Moisture unstable, store under Ar	ChemCatChem 14, e202200961 (2022), Ref. 22
PhS–SiR ₃	Not specified	Low	Rapid solvolysis in dioxane/H ₂ O with trace acid or base, giving thiophenol and silanols	Sulfur Rep. 20, 279–395 (1998), Ref. 18
^t BuMe ₂ SiOCH ₂ CH ₂ SSi ^t BuMe ₂	Moderate	Low	Chromatography sensitive on silica gel, distillation preferred	J. Am. Chem. Soc. 126, 7386–7392 (2004), Ref. 26
	Low	Low	Readily cleaved in IPA/H ₂ O	Eur. J. Org. Chem. 2021, 2694–2700 (2021), Ref. 23
	High, trisiloxane-derived shielding	Low	Partial hydrolysis in IPA/HOAc	Eur. J. Org. Chem. 2021, 2694–2700 (2021), Ref. 23
	Low	Low	Rapid solvolysis to EtOH (< 30 min, rt)	
	Low	High, Branched RSH	Solvolysis in EtOH (16 h, rt)	Dalton Trans., 3401-3411 (2008), Ref. 24
	High	High, Branched RSH	Partial solvolysis in EtOH (50% after 25 h, reflux)	

As shown above, achieving sufficient stability of Si–S linkages has often required the use of specifically substituted or sterically demanding thiol structures in combination with bulky silyl groups. In this context, the key distinction of the present work is that the TTMSS-derived (TMS)₃Si group provides sufficient steric shielding to stabilize the S–Si bond without requiring additional steric demand from the thiol component. This feature allows the formation of stable Si–S linkages across a broad range of thiol substrates and is supported by control experiments (Table S5, SI).

Accordingly, we have revised the sentence in the introduction from “However, in all cases, the Si–S compounds are normally moisture sensitive and easily decompose during storage or column chromatography.¹⁸” to the more cautious and accurate statement: “However, in many reported cases, thiosilanes with traditional silyl protecting groups are normally moisture-sensitive and easily-decomposed during purification on silica gel.^{18, 22, 23, 26}” And Figure 1b has been revised as follows.

(b) Previous works for thiol silylation

2. In Fig. 1b, the authors summarize several previously reported strategies for forming Si–S bonds. Given its relevance, the method described in ref. 29 (Eur. J. Org. Chem. 2021, 2694–2700) should be explicitly incorporated into this summary, together with a brief indication of its advantages/limitations compared to the present TTMSS-mediated protocol.

Response: Description was added as “Brook and coworkers developed a B(C₆F₅)₃-catalyzed hydrosilylation^{23,24} of disulfides using hydrosilanes and silicones” and related information was added in Figure 1b. The reference sequence is changed accordingly: ref. 29 was changed to ref. 23, ref. 30 was changed to ref. 24, and ref. 23~28 was changed to ref. 25~30.

3. In fact, the synthesis of tris(trimethylsilyl)silyl sulfides has already been reported, for example in Tetrahedron 72 (2016) 7764–7769, where (TMS)₃Si–SR type compounds are also accessed. The authors should also cite this reference.

Response: The paper (Tetrahedron 72 (2016) 7764–7769) has now been cited as ref. 31.

4. In Fig. 2b, the yields obtained with different equivalents of TTMSS are shown. When a 1:1 ratio of RSSR to (TMS)₃SiH is used, the reported yield appears to reach ~70%. This is somewhat confusing. In the reaction, RSSR + (TMS)₃SiH → RS–Si(TMS)₃ + RSH, the stoichiometry and the definition of “yield” (with respect to which component?) should be unambiguously consistent.

Response: The yield in Fig. 2 was calculated with respect to RSSR (consistent in all reactions in the manuscript), and indeed, as a “hydrosilylation” reaction, a stoichiometric amount of RSH was generated. Related descriptions were added to clarify this point: “In all cases, thiols were generated as byproducts and the dehydrogenative S–Si coupling between thiol and TTMSS was not observed.”

Referee #3 provided several helpful comments aimed at improving the clarity and rigor of the manuscript.

1. As in the case of cyclic disulfides, I can see why the authors refer to this transformation as hydrosilylation of disulfides. However, Figure 2 makes it look more like a silylation. There is no information shown about the formation of a thiol. Does this mean that, from the disulfide used, one half of the RS groups goes to form a thiosilane while the other half becomes a thiol, or do we obtain only the thiosilane, with S-silylation also occurring? This question is related to another one: if a thiol is indeed formed, how do the authors separate it from the thiosilane? Is this isolation straightforward? The thiols formed would be rather difficult to separate using a rotary evaporator (despite their high volatility), and the process would certainly not be very safe for the personnel.

Response: Indeed, a stoichiometric amount of RSH was generated, and the overall stoichiometry of the reaction can therefore be described as: $\text{RSSR} + \text{R}'_3\text{SiH} \rightarrow \text{RSSiR}'_3 + \text{RSH}$. The formation of the thiol byproduct can be confirmed by crude NMR analysis, as shown in SI, Figure S2. Only the thiosilane was isolated. Efforts have also been made to transform both parts of RSH to thiosilane using oxidants to re-oxidize the thiol back to RSSR; however, the attempts have remained unsuccessful. Related information has been added to Table S2 in the SI. Related descriptions were added in the manuscript as “In all cases, thiols were generated as byproducts and the dehydrogenative S-Si coupling between thiol and TTMSS was not observed. Attempts to re-oxidize the thiol byproduct to the corresponding disulfide for full conversion to the thiosilane were unsuccessful (see Table S2 in the supplementary information for details).”

Table S2. Oxidant Screening^a

Entry	[O]	Yield of 2a (%) ^b	Yield of 3a (%) ^b	s.m. (%) ^b
1	Cu(NO ₃) ₂ ·3H ₂ O	24	0	54
2	Cu(acac) ₂	37	0	39
3	Cu(OTf) ₂	0	0	97
4	CuO	4	0	90
5	Cu(OAc) ₂	28	0	51
6	I ₂	0	5	88
7	CuSO ₄	26	7	32
8	DDQ	0	0	100
9	HFIP	17	24	38
10	S ₈	0	0	95

^aConditions: **1a** (0.10 mmol), TTMSS (2 equiv.), MTBE (0.1 M), [O] (1 equiv.), rt, 12 h. ^b¹H NMR yields using 1,3,5-trimethoxybenzene as the internal standard. Yields are based on RSSR.

2. The authors describe a (accelerated) synthetic procedure consisting simply of mixing the reagents on a rotavap at 35 °C, where the solvent slowly evaporates, ultimately yielding the product. But what happens if this is carried out in a closed system at 35 °C? And what about performing the reaction at this temperature under solvent-free conditions?

Response:

Scheme S1. Comparing sealed-vial, solvent-free, and rotary evaporation conditions.

It is a good question involving the kinetic feature of the reaction. To exclude the influence of the solvent-removal-process, the reaction towards product **2g** was selected as a representative example, and the *in-situ* ^{19}F -NMR was used to detect the kinetic feature of the reaction. As shown above, in comparison to the rapid reaction (99% yield within 5 min.) under rotavap, the conventional closed-system reaction at a concentration of 0.1 M is much slower (88% yield after 24 h). The solvent is also important. Since the substrate is poorly soluble in TTMSS, the solvent-free reaction proceeds more slowly, requiring 30 h to reach 95% yield. Related information is added in Figure 3 and the corresponding description has been added in the manuscript as follows: “Take product **2g** as a representative example, *in-situ* ^{19}F NMR monitoring revealed the kinetic features of the reaction. In contrast to the rapid transformation under rotavap conditions (99% yield within 5 min), the conventional closed-system reaction at a concentration of 0.1 M was much slower (88% yield after 24 h). The solvent-free reaction also proceeded slowly (95% yield after 30 h), probably due to the poor solubility of the substrate in TTMSS.”

3. For the sentence “It is worth noting that the phenol group, normally an active functional group for silylation, was well tolerated in our reaction (product 2h, 96% yield),” it would be useful to refer to some literature. Are there known cases of silylation of OH groups without a catalyst? If so, why is that not the case here? Regardless of the answer (yes or no), this should be backed up with appropriate references (either an original paper or a review).

Response: We thank the reviewer for this helpful suggestion. Indeed, although silicon is an oxyphilic element, silanes are normally stable with phenol without catalysts. Related description has been deleted.

4. Radical trapping by TEMPO. There are known studies showing that TEMPO is not a suitable reagent in the case of thiols. It simply enables the formation of disulfides, in this case regenerating the starting material. It would be worthwhile referring to this literature. Nevertheless, this does provide some confirmation of the proposed mechanism. However, it might be worth attempting a radical clock reaction, for example using aliphatic disulfides bearing a bromo substituent, or disulfides containing cyclopropyl groups.

Response: We thank the reviewer for the comment. The TEMPO experiment, although it was used in some literature (*Org. Biomol. Chem.* **15**, 7678–7684 (2017), was added as ref-50), is indeed not very solid. And we think the radical clock experiment is also difficult in this case, because the stable thiyl radical do not have enough driving force to transform to a carbon radical via ring opening/closure. Instead, an EPR experiment is shown in Figure 6a (TEMPO trapping experiment has been moved from the main text to SI). Under visible-light irradiation (450 nm), a characteristic EPR signal corresponding to a PBN-thiyl radical adduct was observed. A related literature report on PBN trapping of phenylthiyl radicals has also been cited (*Bull. Chem. Soc. Jpn.* **57**, 1745-1749 (1984), $a_{\text{N}} = 13.9$ G, $a_{\text{H}} = 1.8$ G in benzene, was added as ref. 51, and ref. 49~50 was changed to ref. 51~52). The corresponding description has been added in the manuscript as follows: “A spin-trapping experiment with PBN under visible-light

irradiation (450 nm) revealed a characteristic signal of a PBN-thiyl radical adduct upon irradiation, confirming the involvement of sulfur-centered radical intermediates (Fig. 6a).⁵¹

Figure S3. ESR spectra for the spin trapping experiment

Radical **1a-1**: $g = 2.0053$, $a_N = 14.8 \text{ G}$, $a_H = 5.3 \text{ G}$